# Oxidative Potential Induced by Ambient Particulate Matters with Acellular Assays: A Review

**Lanfang Rao [1], Luying Zhang [1], Xingzi Wang [1], Tingting Xie [1], Shumin Zhou [2], Senlin Lu [1,*], Xinchun Liu [3,*], Hui Lu [4], Kai Xiao [5], Weiqian Wang [5] and Qingyue Wang [5,*]**

1. School of Environmental and Chemical Engineering, Shanghai University, Shanghai 200444, China; RaoLanfang@shu.edu.cn (L.R.); zhangluying@shu.edu.cn (L.Z.); xzwang@shu.edu.cn (X.W.); xietingting3683@163.com (T.X.)
2. Lab of Plant Cell Biology, Shanghai Key Laboratory of Bio-Energy Crops, School of Life Sciences, Shanghai University, Shanghai 200444, China; zsm79@shu.edu.cn
3. Institute of Desert Meterorology, China Meteorological Administration, Urumqi 830002, China
4. School of Environmental Sciences, Guangxi Normal University, Guilin 541004, China; luhui1008@163.com
5. School of Science and Engineering, Saitama University, Saitama 338-8570, Japan; xiao.k.662@ms.saitama-u.ac.jp (K.X.); weiqian@mail.saitama-u.ac.jp (W.W.)
* Correspondence: senlinlv@staff.shu.edu.cn (S.L.); liuxch@idm.cn (X.L.); seiyo@mail.saitama-u.ac.jp (Q.W.); Tel.: +86-21-6613-7511 (S.L.); Fax: +86-21-6613-7787 (S.L.)

**Abstract:** Acellular assays of oxidative potential (OP) induced by ambient particulate matters (PMs) are of great significance in screening for toxicity in PMs. In this review, several typical OP measurement techniques, including the respiratory tract lining fluid assay (RTLF), ascorbate depletion assay (AA), dithiothreitol assay (DTT), chemiluminescent reductive acridinium triggering (CRAT), dichlorofluorescin assay (DCFH) and electron paramagnetic/spin resonance assay (EPR/ESR) are discussed and their sensitivity to different PMs species composition, PMs size distribution and seasonality is compared. By comparison, the DTT assay tends to be the preferred method providing a more comprehensive measurement with transition metals and quinones accumulated in the fine PMs fraction. Specific transition metals (i.e., Mn, Cu, Fe) and quinones are found to contribute $OP^{DTT}$ directly whereas the redox properties of PMs species may be changed by the interactions between themselves. The selection of the appropriate OP measurement methods and the accurate analysis of the relationship between the methods and PM components is conducive to epidemiological researches which are related with oxidative stress induced by PMs exposure.

**Keywords:** ambient particles; oxidative potential; acellular assay; transition metals; quinones

## 1. Introduction

Numerous epidemiological and experimental studies between exposure to ambient particulate matters (PMs) and lots of adverse health effects such as respiratory and cardiovascular diseases and diabetes have been carried out [1–5]. Compared to other air pollutants, PMs are complex in chemical composition and source origination, and their capability to absorb large amounts of toxic chemicals [6,7]. Adverse health effects which are most potentially caused by PMs are thought to be PMs-induced oxidative activity. Reactive oxygen species (ROS) are produced when PMs interact with epithelial cells and macrophages.

There are two kinds of ways for PMs to induce ROS in the human body: (1) oxidants existing on and/or within the particle itself are deposited in respiratory systems; (2) cells are stimulated by certain chemicals in PMs to produce excess ROS or specific biochemicals interact with components in PMs to

produce ROS [8,9]. ROS are highly reactive due to their unpaired electrons, and they include hydroxyl radical (●OH), hydrogen peroxide ($H_2O_2$), organic peroxyl radicals ($RO_2$), superoxide radical ($O_2^{\bullet-}$) and hypochlorite ion ($OCl^-$) [9].

　　Previous studies mostly defined "oxidative potential" (OP) as the generation of ROS and depletion of antioxidants catalyzed by inhaled PMs [9,10]. Oxidative potential, representing the capacity of PMs to oxidize molecules with the generation of ROS, can be used as a plausible metric of PMs toxicity [11]. Compared to the cellular assays, acellular assays have the advantage of faster reading speed, lower price, less control environments and being suitable for automation. Each acellular OP assay has certain specificity for the ROS or the precise type of ROS-inducers, leading to the fact that none of the methods has been used as a standard method to assess toxicity of ambient particles [12].

　　Considering that different assays focus on different chemical fractions of the oxidative activity caused by the PMs, there are no clear criteria for which OP assay and results are likely to be the best [13]. This work briefly describes some commonly adopted acellular OP assays, including electron paramagnetic/spin resonance (ESR assay, $OP^{ESR}$), dithiothreitol assay (DTT assay, $OP^{DTT}$), dichlorofluorescin assay (DCFH assay, $OP^{DCFH}$), ascorbic acid assay (AA assay, $OP^{AA-only}$) and the respiratory tract lining fluid assay (RTLF assay, $OP^{RTLF}$) firstly. Their sensitivity to different PMs species composition, PMs size distribution and seasonality are discussed in the following part with the aim of providing a preferred method to measure OP. Because of the comprehensiveness of the DTT assay, its driving factors are introduced as the focus in this review. Understanding the differences between acellular OP techniques, as well as the varying relationship to PMs composition and size distribution, is conductive to future research on investigating the relevance between epidemiologic disease and OP assay.

## 2. Oxidative Potential Measurement Methods

　　Oxidative potential is considered as a reasonable indicator of PMs toxicity. Some acellular methodologies are used to quantify the OP and demonstrate the complex mechanisms of the generation of ROS. Some assays measure the OP of PMs by the loss of a proxy of cellular reductants (i.e., $OP^{DTT}$), or endogenous antioxidant species (i.e., RTLF assay, AA assay). ESR assay measures the generation of hydroxyl radical in the presence of $H_2O_2$. Fluorescence intensity in DCFH assay and the intensity of the emitting light in CRAT assay are both converted into equivalent $H_2O_2$ concentrations. In order to ensure the standardization of interlaboratory measurement, residual oil fly ash (ROFA) always is selected as a positive control and inert carbon black as a negative control. Blank filters are also routinely extracted and run through assays system [14–17].

### 2.1. Respiratory Tract Lining Fluid Assay

　　The respiratory tract lining fluid (RTLF) represents the first detoxifying environment encountered by inspired particulate matter [18] and has been shown to contain both small molecular weight and high concentrations of the antioxidants uric acid (UA) and ascorbic acid (AA), and reduced glutathione (GSH) [14]. The structure of urate and its initial degradation products can be seen in Figure 1. GSH is a tripeptide and its reaction with reactive species in PMs can often generate thiyl ($GS^{\bullet}$) radicals. Any available $GS^-$ can react quickly with $GS^{\bullet}$ to form glutathione radical disulfide anion ($GSSG^{\bullet-}$) and in turn $O_2^{\bullet-}$ (Equation (3)). This assay detects concentration variation of antioxidant molecules in the simplified synthetic human respiratory tract lining liquid after mixing with particle suspension [17]. PMs sample and composite antioxidant solution (200 μM) pH were adjusted to pH 7.0. The mixture was then transferred into an incubator maintained at 37 °C for 4 h [14]. Antioxidant concentrations were determined after incubations by high-pressure liquid chromatography (HPLC) and glutathione-reductase enzyme recycling method respectively [19]. The extent to which PMs

depleted antioxidants provided not only a quantitative PMs oxidative potential, but also reflected reactions that may occur at the air-lung interface in the body.

$$GSH \Leftrightarrow H^+ + GS^- \tag{1}$$

$$GS^{\bullet} + GS^- = GSSG^{\bullet-} \tag{2}$$

$$O_2 + GSSG^{\bullet-} = GSSH + O_2^{\bullet-} \tag{3}$$

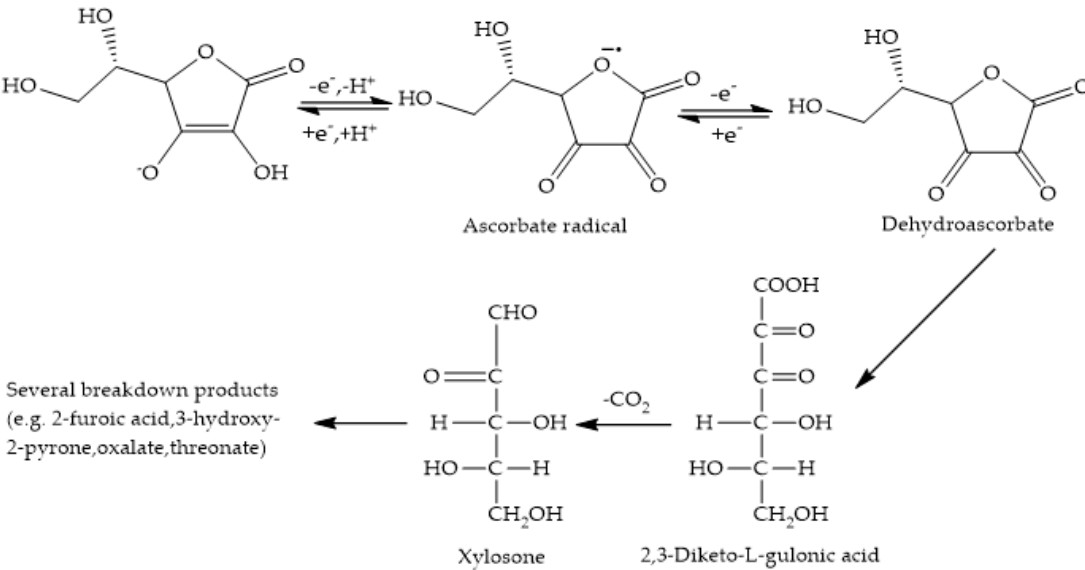

**Figure 1.** Urate can be oxidized by urate oxidase and forms 5-hydroxyisourate (HIU) initially [20].

The Ascorbate Depletion Assay

In the ascorbate (AA) depletion assay, only ascorbate acid is used which is thought to be a simplified version of the RTLF assay [21]. AA is oxidized to dehydroascorbic acid with the decrease in redox active substances in PMs. Figure 2 shows the structure of ascorbate and its oxidation and degradation products. In the AA assay, PMs extracts were incubated at 37 °C, after adding ascorbate acid. The reaction occurred directly, and was then measured in a UV/VIS-spectrophotometer with spectral scans (190–350 nm) performed at intervals of 10 min [13,22]. The depletion rate of AA was applied to represent the PM oxidative potential.

**Figure 2.** Structure of ascorbate and its oxidation and degradation products [23].

## 2.2. Dithiothreitol Assay

The dithiothreitol (DTT) could reduce oxygen to the superoxide anion ($O_2^{\bullet-}$) that could then form other ROS (i.e., $\bullet$OH). Redox active compounds oxidized DTT to its disulfide form and transferred

an electron to $O_2$, generating $O_2^{\bullet-}$ (Figure 3) [8,24]. In the DTT assay, particles were first extracted using Milli-Q or methanol. The PMs extraction solution was then incubated for a period with DTT in a phosphate buffer. At preset times, an aliquot removed from the mixture was added with trichloroacetic acid (TCA) (quench reaction), Ethylene Diamine Tetraacetic Acid (EDTA) and 5,5′-dithiobis-(2-nitrobenzoic acid) (DTNB) solution. The reaction of residual DTT and DTNB formed 2-nitro-5-thiobenzoic acid (TNB), which could be quantified by a UV/VIS spectrophotometer at 412 nm [21,25,26]. When DTT was in excess, the consumption rate of DTT which was dependent on the linear slope of DTT depletion was proportional to the concentration of redox-active species in the PMs sample. The OP responses could be normalized by the volume of sampled air as an exposure metrics or by the PMs mass to represent the intrinsic ability of the PMs to deplete relevant antioxidants.

**Figure 3.** Reactions between dithiothreitol (DTT) and $O_2$ with particulate matters (PMs) as a catalyst [27].

### 2.3. Chemiluminescent Reductive Acridinium Triggering Assay

Chemiluminescence of acridine esters under alkaline conditions is the foundation of the chemiluminescent reductive acridinium triggering (CRAT) assay [28]. The mechanism of this reaction can be seen in Figure 4. DTT or GSH were adopted as reductants in the CRAT assay to produce $H_2O_2$, which reacted with acridinium ester after adding a slightly alkaline buffer into the solution. The emitting light during this reaction was determined by about 1 s with a luminescence meter. The intensity of the emitting light could be applied to quantify rates of $H_2O_2$ production [21]. CRAT, a relatively new methodology, has not yet been widely used in large air pollution studies.

**Figure 4.** Mechanism for the chemiluminescence reactions of chemiluminescent reductive acridinium triggering (CRAT) assay [28].

### 2.4. Dichlorofluorescin Assay

Dichlorofluorescin is a non-fluorescent reagent that fluoresces when oxidized [29]. In the presence of horseradish peroxidase (HRP), DCFH could be rapidly oxidized to a fluorescent compound (DCF) (Figure 5). Prior to analysis, the DCFH in sodium phosphate buffer (pH = 7.2) was mixed with horseradish peroxidase (HRP) in a ratio [30]. This experiment was performed under dark conditions

to prevent DCFH photooxidation and reduce the variability of background $H_2O_2$ concentration [31]. Sample filters were suspended in a beaker containing DCFH–HRP, and sonicated to extract ROS from the particles. The formed DCF was determined by fluorescence at the excitation wavelengths of 485 nm and emission wavelengths of 530 nm. The measured fluorescence intensities were converted into equivalent $H_2O_2$ concentrations using least-squares analysis with a $H_2O_2$ calibration curve [29,32,33].

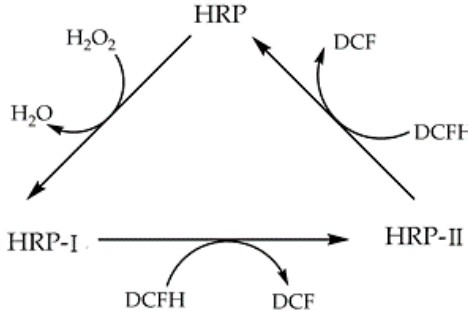

**Figure 5.** $H_2O_2$ reacts with horseradish peroxidase (HRP), following the reaction with dichlorofluorescin (DCFH) to produce fluorescence product fluorescent compound (DCF) [31].

## 2.5. Electron Paramagnetic/Spin Resonance Assay

The electron paramagnetic/spin resonance (EPR/ESR) assay was developed for the direct detection and quantification of materials containing unpaired electrons, such as free radicals or transition metal ions [13,34,35]. The physical assay can be measured by free radicals from the particles collected on filters. Stronger free radical signals generated from coal combustion fine particles (PM < 1.1 μm) could be measured (Figure 6).

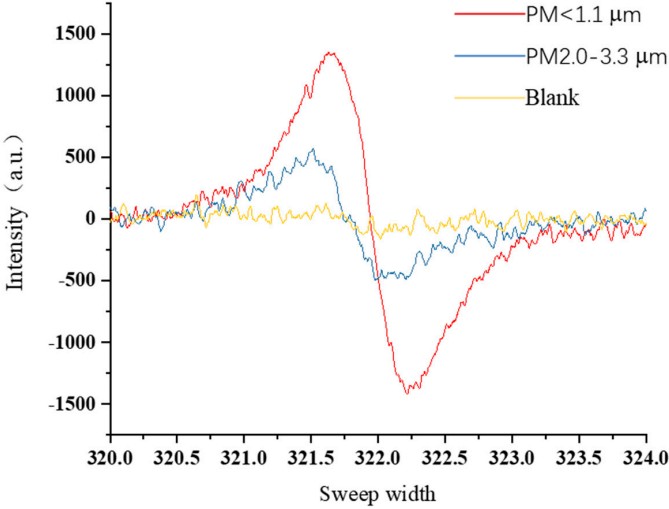

**Figure 6.** Free radicals generated from coal combustion particles. Free radical intensity varied with different particle sizes. Fine particles (PM < 1.1 μm) could generate stronger free radicals (present study, not published).

Additionally, the ESR assay could be used to test different free radicals with different spin traps, and the free radical spectrum pattern was different with different spin traps. Four peaks could be found in the pattern with 5,5-dimethyl-1-pyrroline-N-oxide (DMPO) spin trap, while only three peaks showed with 1-hydroxy-2,2,6,6-tetramethyl-4-oxo-piperidine (TEMPONE-H) (Figure 7). Oxidative potential of particles on filters based on the capacity of PMs to generate •OH via Fenton-type reaction in the presence of $H_2O_2$ with 5,5-dimethyl-1-pyrroline-N-oxide (DMPO) as a specific spin-trap [36,37].

DMPO–OH adducts could be formed from direct trapping of ●OH (Equation (4)) or the decomposition of DMPO–OOH (Equation (5)) [36]. The OP was calculated from the sum of total amplitudes of the DMPO–OH quartet signal and expressed as the total amplitude arbitrary units divided by the volume of sampled air or weight of PMs [38,39].

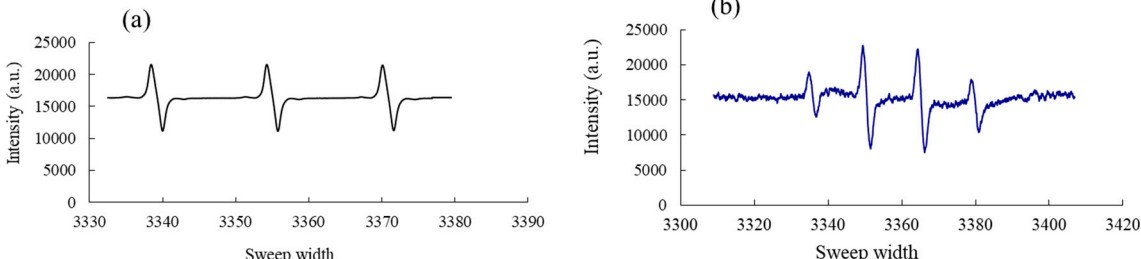

**Figure 7.** ESR spectra showed different patterns with different spin traps: (**a**) 1-hydroxy-2,2,6,6-tetramethyl-4-oxo-piperidine (TEMPONE-H) and (**b**) 5,5-dimethyl-1-pyrroline -N-oxide (DMPO) (present study, not published).

In addition to DMPO, another spin-trap 1-hydroxy-2,2,6,6-tetramethyl-4-oxo-piperidine (TEMPONE-H) was adapted to quantify total concentrations of ROS (Figure 7). TEMPONE-H is a closed shell molecule that could be transformed to radicals by deprotonation in a reaction with radicals. TEMPONE-H reduces peroxynitrite to $NO_2$ (Equation (6)) and reduces $NO_2$ to nitrite (Equation (7)). Moreover, TEMPONE-H can react with superoxide radicals (Equation (8)) and peroxy radicals (Equation (9)) with a formation of 1-hydroxy-2,2,6,6-tetramethyl-4-oxo-piperidinoxyl (TEMPONE) [40,41].

$$DMPO + \bullet OH = DMPO\text{-}OH \tag{4}$$

$$DMPO + O_2{}^{\bullet-} \rightarrow DMPO\text{-}OOH \rightarrow DMPO\text{-}OH \tag{5}$$

$$TEMPONE\text{-}H + ONOOH = TEMPONE + NO_2 + H_2O \tag{6}$$

$$TEMPONE\text{-}H + NO_2 = TEMPONE + HNO_2 \tag{7}$$

$$TEMPONE\text{-}H + HO_2{}^- = TEMPONE + H_2O_2 \tag{8}$$

$$TEMPONE\text{-}H + RO_2 = TEMPONE + RO_2H \tag{9}$$

## 3. Comparison of Acellular Assays

Chemical constituents of PMs drive different responses by different assay types, and particle sizes also have significant influences on OP outcomes [9]. In order to provide a preferred method to measure OP, the sensitivity to different PMs species composition, size distribution and seasonality are discussed.

### 3.1. Sensitivity of Different Acellular Assays

Different OP assays capture different ROS species due to different redox reactions [9]. The DCFH assay measures particle-bound ROS that are inherently existing on and/or within the particle itself [29]. $OP^{ESR}$ measures the certain production of ROS over time whereas $OP^{RTLF}$ and $OP^{AA}$ measure the depletion of target molecules [28].

Certain OP assays respond differently by panels of chemicals and may also vary due to interactions between different components in PMs. For example, the DTT assay has different responses to metals and organic compounds. The $OP^{DTT}$ was also sensitive to organic compounds (i.e., water-soluble organic carbon (WSOC), elemental carbon (EC), organic carbon (OC), black carbon, quinones and hydroxyquinones). $OP^{CRAT}$ could react to trace metals Fe or Cu and quinones too [42,43]. $OP^{AA\text{-}only}$ was almost exclusively related with Cu(II) [44]. In the RTLF assay, $OP^{AA}$ and $OP^{GSH}$ responded to different components of PMs as they were not significantly correlated with one another [45]. $OP^{GSH}$

was uniquely correlated with water-soluble Cu [19] and OP$^{AA}$ had a strong positive correlation with Fe and Cu, the main elements that track non-exhaust traffic emissions [46]. Interestingly, OP$^{AA-only}$ had a much stronger correlation with Cu(II) than OP$^{AA}$ in the RTLF assay [44]. The OP$^{ESR}$ was highly correlated with the traffic-related PMs components (i.e., Fe, Cu). OP$^{ESR}$ and OP$^{AA}$ showed the most similar results due to their large metal dependences [42].

### 3.2. PMs Size Distribution

Generally, redox active species with PMs size indicate that the OP of PMs is size determined [24]. OP$^{DTT}$ in PM$_{2.5}$ was significantly higher than that in PM$_{10}$ on a per mass basis [42] and per volume unit [47]. But sometimes, it was the reverse case for OP$^{ESR}$ [42]. The OP$^{ESR}$ in PM$_{10}$ was 4.6 times greater than that in PM$_{2.5}$ when expressed per volume unit, and 3.1 times greater when expressed per mass unit showing higher OP in larger PMs size fractions. OP$^{AA}$ was also more affected by components in coarse PMs as its responses were significantly higher for PM$_{10}$ than that for PM$_{2.5}$ [48].

However, for a more refined division of particle size, OP$^{DTT}$ was unimodal (Figure 8). Volume-normalized OP$^{DTT}$ and mass-normalized OP$^{DTT}$ peaked at submicron 0.56−1 μm and 0.1−0.32 μm respectively [24]. OP$^{DTT}$ and OP$^{DCFH}$ were similar in distribution, peaking at 0.32–1.8 μm, but AA assay was completely different. OP$^{AA}$ was almost exclusively sensitive to PMs in the coarse mode (3.2–5.6 μm) [49]. The particles from brake/tire wear containing very high concentrations of Cu, Fe and Mn also belonged to the coarse fraction in the range of 3.2–5.6 μm. The similar distribution sizes indicated that OP$^{AA}$ was significantly sensitive to PMs in the coarse mode produced by vehicular traffic, such as brake wear and re-suspended road dust. OP$^{DTT}$ and OP$^{DCFH}$ were more sensitive to fine PMs generated by combustion processes [49]. In the study of Fang et al., (2017), water-soluble OP$^{DTT}$ peaked near 1−2.5 μm whereas water-insoluble OP$^{DTT}$ was bimodal with both fine and coarse fractions due to the absorbed reactive species on the surfaces of PMs [50]. Overall, it seems that ultrafine and fine PMs are more important to OP$^{DTT}$ value, but more work is required to reach precise conclusions.

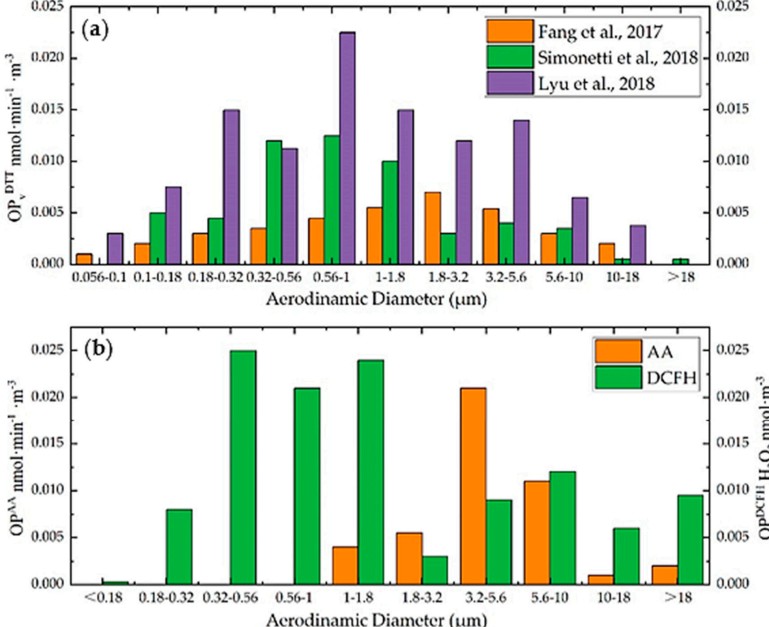

**Figure 8.** Oxidative potential of water-soluble fraction extracted from different size particles on a per volume unit. (**a**) the highest level of the oxidative potential (OP) was found in the size range from 1−2.5 μm, 0.32–1.8 μm and 0.56–1 μm by using DTT assay respectively [24,49,50]; (**b**) OP$^{AA}$ was almost sensitive to particles in the coarse mode (3.2–5.6 μm) and OP$^{DCFH}$ peaked at 0.32–1.8 μm [49].

### 3.3. OP Related with the Chemical Composition of PMs Collected in Different Seasons

Generally, significantly higher OP response could be measured in winter than that in summer for DTT and AA assays. The volume- and mass-based DTT (DTTv and DTTm) values of $PM_{2.5}$ were significantly higher in the cold season possibly driven by biomass burning emission seasonality [51–53]. Average $OP^{DTT}$ was 1.5 times and $OP^{AA}$ was 2.3 times higher in the cold season than in the warm season [48]. However, $OP^{AA}$ reached highest values near heavily trafficked highways and $OP^{DTT}$ was more spatially homogeneous [44]. Calas et al., (2018) reported seasonal variations of the OP value by using four acellular assays (DTT, AA, RTLF and ESR) over a 1-year period, and showed only $OP^{ESR}$ did not show seasonality and showed weaker relationships with other assays and chemical species [54]. But sometimes the OP data are different. In an urban background area of Athens, Greece, the OP value that used the DCFH assay was higher in summer, mainly as a result of higher concentrations of EC and WSOC during the warm season [55]. In Beijing, the seasonal averages of $OP^{DTT}$ exhibited peak values in summer. This was correlated with the higher concentration of water-soluble organic components in summer produced by the photochemical formation of secondary organic aerosols [56].

### 3.4. Correlation with Health Impacts

Among the studies about the relationship between epidemiology and OP of PMs, the $OP^{DTT}$ assay was a main method. $OP^{DTT}$ has been found to be linked with several biological end points, such as, fraction of nitric oxide (FeNO) in exhaled breath [57], and mammalian cell cytotoxicity of $PM_{2.5}$ [58], increased relative risk for asthma [5] and congestive heart failure [59]. A study in the Netherlands found that asthma and rhinitis were positively related with $OP^{DTT}$, but not significantly associated with $OP^{ESR}$ [60]. Maikawa et al., (2016) investigated the relationship between airway inflammation in asthmatic children (FeNO) and oxidative potential of $PM_{2.5}$ using RTLF assay. A positive correlation was found between FeNO and $OP^{GSH}$ but not found between $PM_{mass}$ or $OP^{AA}$ [61]. $OP^{DTT}$ and $OP^{GSH}$ are the most relevant to health among several OP assays [9,13].

From the above, the DTT assay tends to be the preferred method to evaluate the oxidative potential, providing a more comprehensive measurement due to its high sensitivity to transition metals and organic compounds, which are mainly accumulated in fine PMs fractions (Table 1) [10]. DTT assay is highly correlated with health effects to better understand the link between PMs and human health. The DTT approach is the major point of discussion in the next section in this review in terms of driving forces.

**Table 1.** Comparison of acellular assays to assess the oxidative potential.

| Assay | Requirements | Estimated Way | Sensitivity | Sources | Characteristic | Reference |
|---|---|---|---|---|---|---|
| DTT | fast, inexpensive, easy to perform and suitable for automation | the depletion rate of chemical proxies for cellular reductants | organic compounds; traffic-related metals; inorganic ions | biomass burning; brake/tire wear; traffic/fossil fuel combustion; photochemical aging | associated with fine fraction | [24,42,47,50] |
| AA | | the antioxidant loss rate | metals | non-exhaust traffic emissions | associated with coarse particles | [48,49] |
| GSH | | the antioxidant loss rate | Cu | non-exhaust traffic emissions | not a strong marker for traffic | [43] |
| DCFH | | the increase in fluorescence intensity over time | organic compounds; inorganic ions | anthropic combustion; secondary aerosol | associated with fine fraction | [49] |
| CRAT | has not been widely used | the chemiluminescence reaction | transition metals, quinones | ambient particles | highly correlated with PM mass concentration | [21] |
| ESR | relatively little material, inexpensive | the ability of PMs to generate •OH | transition metals; organic components | ambient particles | associated with coarse/fine particles | [42] |

## 4. Drivers of Oxidative Potential

Considering that the PMs is consisted of organic and inorganic matters, these components could exacerbate free radical reaction and contribute to the measured OP response, and therefore the relationship between the OP and the main chemical components would play a key role in understanding the health risk (Table 2). In general, these following inorganic and organic species in the PMs showed the closest relationships with the OP.

### 4.1. Trace Metals

Metals, constituted 6–13% of the PMs mass [62], have been characterized in detail such as Ca, Mg, Ba, Al which originate from resuspension of soil dust emissions [49,63,64], while Fe, Cu, Zn, Cr, Cd are the transition metals associated with non-exhaust traffic emissions [62]. K could be considered as tracers of the biomass burning [33,48]

$$M^{n+1} + R(SH)_2 = M^n + RSHS^{\bullet} + H^+ \tag{10}$$

$$RSHS^{\bullet} + O_2 = RSS + H^+ + O_2^{\bullet-} \tag{11}$$

$$2O_2^{\bullet-} + 2H^+ = H_2O_2 + O_2 \tag{12}$$

$$M^n + H_2O_2 = M^{n+1} + HO\bullet + OH^- \tag{13}$$

$$M^{n+1} + O_2^{\bullet-} = M^n + O_2 \tag{14}$$

Transition metals (M, above) which have a certain amount of oxidative potential can catalyze the oxidation of DTT easily in Equation (10) [65]. The superoxide anion is formatted by the reaction between electron donors (DTT) and molecular oxygen (Equation (11)) [66]. Taking transition metals Cu and Fe for example, Cu(I) and Fe(II) can generate the highly reactive oxidant hydroxyl radical via the Fenton reaction (Equation (13)). And Cu(II) can be reduced to Cu(I), and Fe(III) to Fe(II)) to complete a redox cycle (Equation (14)).

**Table 2.** Summary of driving species of oxidative potential (OP) mainly measured by DTT assay reported in literatures.

| Location | Particles | Seasons | Sampling Period | Assay | Driving Species | Reference |
|---|---|---|---|---|---|---|
| Atlanta | $PM_{2.5}$ | One year | Jan–Dec, 2017 | DTT | BrC, EC, K, Fe, Cu | [51] |
| Central Mediterranean Sea | $PM_{10}$, $PM_{2.5}$ | One year | 2014-2015 | DTT, AA | $K^+$, $NO_3^-$, Ba, Cd, Cu, Fe, Mn, P, V, OC, EC | [48] |
| Atlanta | $PM_{2.5}$ | One year | 2017 | DTT, RTLF | WSOC, OC, EC, Fe, Cu, Mn | [19] |
| the University of Illinois, Urbana−Champaign | Ambient $PM_{2.5}$ | Spring | Feb–Apr, 2017 | DTT | HULIS, Fe, Cu, Mn | [67] |
| the Central Mediterranean basin | $PM_{10}$ | One year | Dec 2014–Oct 2015 | DTT AA | Ba, Cd, Ce, Cr, Cu, Fe, EC, OC | [68] |
| Indo-Gangetic Plain | $PM_{2.5}$ | Winter | 2014 | DTT | OC, EC, WSOC, | [64] |
| Beijing | $PM_{2.5}$ | One year | 2012 | DTT | HULIS | [69] |
| Italy | $PM_{10}$ | One year | Feb–Nov 2015 Apr–May 2016 | DTT, AA | $SO_4^{2-}$, $NH_4^+$, $K^+$, $Mg^{2+}$, $Ca^{2+}$, Ca, Mg, K, Mn, Cu, Rb, Zn, WSOC | [70] |
| the littoral zone of the Bohai Sea | $PM_{2.5}$ | One year | 2016 | DTT | WSOC, EC, Mn, Co, Fe, Cr, Cd, $SO_4^{2-}$, $NH_4^+$, $NO_3^-$ | [52] |
| Italy | Size-segregated PMs | spring | Feb–Mar, 2017 | AA, DTT, DCFH | Cu, Fe, Mn, As, B, Cd, Cr, Mo, Se, Ni, Pb, K, Rb | [49] |
| the Los Angeles Basin | $PM_{2.5}$ | winter | Oct 2014–Jan 2015; Nov 2015–Jan 2016 | DTT | Ba, Cr, Cu, Mn, Ni, Pb, Sb and Zn, EC, OC | [71] |

Abbreviations: BrC: brown carbon; EC: elemental carbon; OC: organic carbon; HULIS: Humic-like Substances, WSOC: water-soluble organic carbon.

Numerous studies have assessed the intrinsic OP induced by metals [24,27,72,73]. A summary of studies is presented in Figure 9. In this review, we followed the procedure of Lu et al., (2019) for the DTT assay [74]. Interestingly, the units of this review are different from other literatures, but the underlying trends of the intrinsic OP of quinones and metals are the same. As shown in Figure 9, Cu(II) and Mn(II) were the most active metals in the DTT assay and Co was the third most reactive metal. The reactivity of other transition metals followed the pattern V(V) ≈ Ni(II) > Pb(II) ≈ Fe(II) > Fe(III). In Figure 10, Cu(II) and Mn(II) were well fitted by power functions, but the concentration responses for Fe(II) and Fe(III) were linear [24]. Even though the intrinsic OP of metals have been quantified by using inorganic salt solutions, the importance for OP assessment of ambient PMs also depends upon the mass concentration of each metal. For example, Fe which made a modest contribution on $OP^{DTT}$ values accounted for the majority of DTT loss from typical ambient $PM_{2.5}$ due to its abundant concentrations [24,73]. Therefore, Fe was responsible for the DTT assumption. However, the solubility of metal ions can be enhanced by the presence of organic ligands in the PMs. For example, the solubility of Fe was strongly correlated with the concentration of oxalate [75]. Sulfate was an important proxy for Fe solubility by affecting aerosol pH [62].

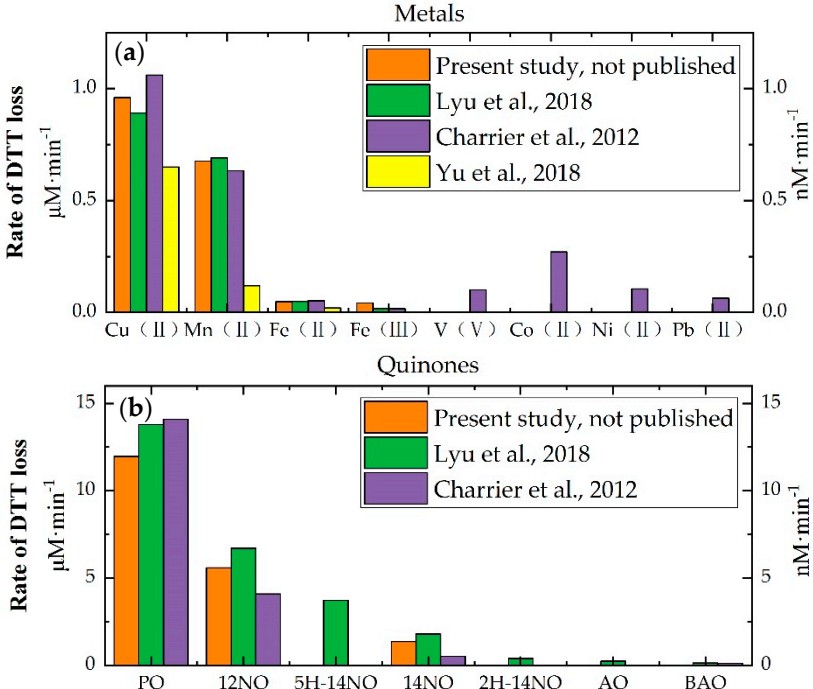

**Figure 9.** Intrinsic OP of individual (**a**) metals and (**b**) quinones [24,67,73]. The concentration of each species was 1 μM. The data in this paper are in nM·min$^{-1}$ while in other literatures they are in μM·min$^{-1}$.

The concentration of H$^+$ could easily destroy the mineral structures to release the structural or interlayer Fe [76]. The dissolved Fe from minerals (e.g., pyrite, illite, chlorite or kaolinite) had considerable oxidative potential [76]. Moreover, minerals could induce ROS formation as a "carrier" of more toxic species such as metals and surface-absorbed polycyclic aromatic hydrocarbons (PAHs). Besides, Fe and Cu were very efficient in forming complexes with Suwanee river fulvic acid (SRFA), making them unlikely to exist in free ionic forms and significantly altering their OP [72].

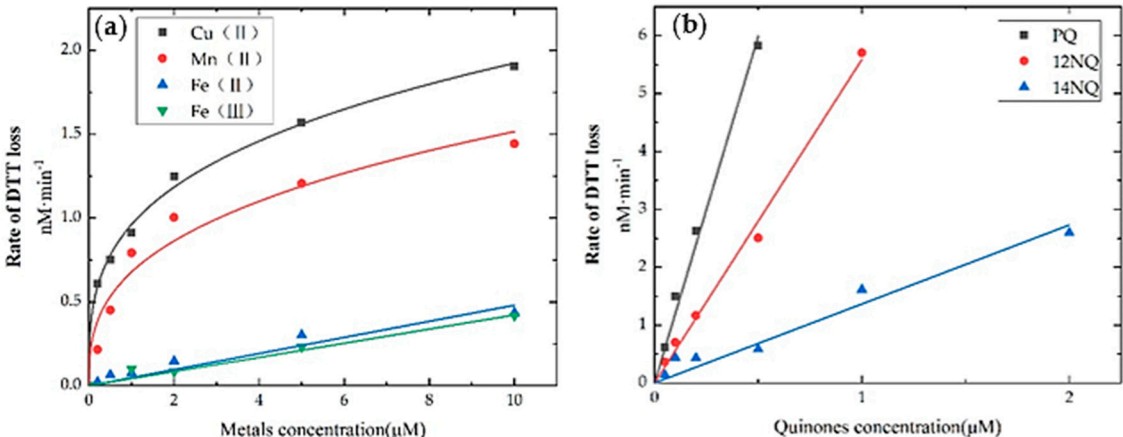

**Figure 10.** Rate of DTT loss as a function of concentration of redox-active species: (**a**) results for the four metals that oxidized DTT; (**b**) results for the three DTT-reactive quinones (present study, not published).

The correlations between $OP^{DTT}$ and metal concentration in ambient PMs do not indicate causation, since particle species tend to be highly covariate. In winter, the strong correlation with non-redox active metal K should be analyzed carefully because this association may be due to a similar increase in K and quinones concentrations, rather than establishing an inevitable causal relationship [77]. The redox inert ions (i.e., Zn, Cd, Pb) also contribute to the oxidative stress of PMs exposure by interacting with cysteinyl thiols on glutathione and key regulatory proteins [66,78]. And $Ca^{2+}$, $Mg^{2+}$, potentially presenting in large quantities in $PM_{2.5}$, can strongly modulate PMs acidity [79].

### 4.2. Carbonaceous Species

Carbonaceous species include elemental carbon (EC), organic carbon (OC) and water soluble organic carbon (WSOC). OC may be widely used as a tracer of biomass burning [80]; polycyclic aromatic hydrocarbons (PAHs) are emitted during incomplete combustion processes including traffic [55]; and quinones are the markers of photochemical formation of secondary organic aerosols (SOA) [33].

Quinones are capable of destroying the proximal thiol of DTT [65,81]. Taking the reaction between DTT and 14NQ for example in the Figure 11, DTT has two thiol groups in close proximity that readily form a stable six membered ring with 14NQ reduced (Reaction 1). And then 14NQ rearranges to form the 1,4-naphthoquinol (Reaction 2). Quinols form a free radical-semiquinone through two ways (Reaction $3_a$, $3_b$). Then the semiquinone is transformed to the original quinone by reacting with $O_2$ to complete a redox cycle (Reaction 4). Similarly, many researchers started with the standard solutions of known quinones (9,10-phenanthraquinone, 1,2-naphthoquinone, 1,4-naphthoquinone, 2hydroxy-1,4-naphthoquinone, 9,10-anthracenequinone, benz(a)anthracene-7,12-quinone and 5-hydroxy-1,4-naphthoquinone) measuring the oxidation potential of quinones (Figure 9) [67]. As shown in Figure 10, the rates of DTT consumption from quinones were proportional to their concentration [24]. PQ was the most reactive specie, followed by 12NQ, 5H-14NQ, 14NQ, 2H-14NQ, AQ, and BAQ. On a concentration normalized basis, quinones, especially PQ, could be much more reactive than other species, but their concentrations were generally low [73]. Quinones in traffic-related PMs, especially NQ and AQ, had considerable pro-inflammatory and genotoxic potential which could cause lung impairment. As the secondary derivates of PAHs in the air, quinones made a great contribution to the toxicity of airborne PMs [82]. There exists correlation between mass level of PAHs, quinones and DTT consumption [73]. Quinones in the atmosphere are formed by tropospheric conversions of precursors PAHs via photochemical reactions and reactions with •OH, nitrate radicals and ozone ($O_3$) [24,83,84]. Many studies showed a correlation between OC and/or WSOC and $OP^{DTT}$ of PMs collected during different seasons [63,64,74,85]. Besides, Chen et al.,

(2019) found that the light absorbing substance BrC3 and the fluorescent substance C7 were important contributors to the DTT activity of water-soluble PMs [8].

**Figure 11.** The reaction mechanism between DTT and 1,4-naphthoquinone (14NQ), including 1, 2, 3a, 3b and 4. Reaction 1 shows DTT readily forms a stable six membered ring with 14NQ reduced; Reaction 2 shows 14NQ rearranges to form the 1,4-naphthoquinol; Quinols form a free radical-semiquinone through Reaction 3a or 3b and the semiquinone is transformed to the original quinone by Reaction 4 [65].

Biomass burning which could release high levels of particulate matter with PAHs and volatile organic compounds (VOCs) played a primary role in PMs capability to generate ROS [49,86]. And there was also a large amount of quinones in fresh wood smoke organic aerosols, which could well explain the DTT consumption [87]. In the meantime, $OP^{DTT}$ caused by vehicle emission particles was high because of the semivolatile organic species [85].

There could be both synergistic and antagonistic interactions among PMs components, which could cause different OP responses [88]. Humic-like substances (HULIS) are water soluble compounds [89]. HULIS, consisting of powerful chelating functional groups (i.e., carboxyl (-COOH), hydroxyl (-CH$_2$OH) and carbonyl (-COCH$_3$), can chelate the transition metals in aerosols and are capable of participating in redox cycling. Higher $OP^{DTT}$ value was been found in HULIS-Fe(II) complexes than Fe(II) alone [74]. Fe(II) itself can produce ROS species through Fenton reactions (Equation (13)), and Fenton-based HULIS-Fe(II) complex system can be expressed through Equations (15)–(17). Quinones could catalyze the generation of H$_2$O$_2$ and •OH [27]. And the interactions between quinones and Fe were additive in $OP^{DTT}$ but synergistically in forming •OH. Magnesium could have synergistic effects with quinones, but Cu were different in the $OP^{DTT}$ assay [67]. When mixing with Cu, the $OP^{DTT}$ value of AQ decreased significantly [88]. And there could be synergistic effects on •OH generation for the mixtures of Cu(II) + Fe(II), PQ + Fe(II), and Cu(II) + Fe(II) +1,2-NQ, Cu(II) + Fe(II) + PQ [90]. Fe(II) should be rapidly oxidized to Fe(III) (Equation (13) (18)), and Cu(I) can also readily reduce Fe(III) (Equation (19)). Cu(II) is continuously reduced by $O_2^{\bullet-}$, providing a steady source of reduced iron (Equation (14)) [91]. Thist may explain the synergistic effect of Fe(II) and Cu(II).

$$\text{Fe(II)} + \text{HULIS} = \text{HULIS-Fe(II)} \tag{15}$$

$$\text{HULIS-Fe(II)} + O_2 = \text{HULIS-Fe(III)} + O_2{}^{\bullet-} \qquad (16)$$

$$\text{HULIS-Fe(II)} + H_2O_2 = \text{HULIS-Fe(III)} + HO\bullet + HO^- \qquad (17)$$

$$\text{Fe(II)} + O_2 = \text{Fe(III)} + O_2{}^{\bullet-} \qquad (18)$$

$$\text{Fe(III)} + \text{Cu(I)} = \text{Fe(II)} + \text{Cu(II)} \qquad (19)$$

*4.3. Ionic Species*

$K^+$, $Cl^-$, $Na^+$, $SO_4{}^{2-}$, $NO_3{}^-$, $NH_4{}^+$ are the main ions in the PMs [63,92]. These ions also played roles in assessment of the OP. According to Patel et al., (2018), there was a significant negative correlation ($p < 0.05$) with $SO_4{}^{2-}$, $NO_3{}^-$, $NH_4{}^+$ which indicated that they do not affect DTT activity [64]. But $SO_2$ and $NO_2$ which are the main precursors of secondary inorganic aerosols showed strong correlations with DTT [63]. Secondary acids, such as ammonium sulfate, can have an indirect effect on OP through reducing pH in the assay of $OP^{DTT}$ [62,76].

*4.4. Water-Insoluble Components*

It was noted that insoluble fraction of PMs had a relationship with the OP. The $OP^{DTT}$ values of ambient particles without filtration were higher than that of water-soluble fraction [93]. In the studies of Gao et al., (2020) and Fang et al., (2017), water-insoluble OP ($OP^{WI\text{-}DTT}$) was determined by the difference between the water-soluble ($OP^{WS\text{-}DTT}$) and total OP ($OP^{Total\text{-}DTT}$). The $OP^{WI\text{-}DTT}$ comprised 20% of total PMs OP on average [50,51]. However, the main drives of $OP^{WS\text{-}DTT}$ and $OP^{WI\text{-}DTT}$ appeared to be the same, implying that there is a connection between $OP^{WI\text{-}DTT}$ and dust surface property.

## 5. Conclusions

Acellular assays including the RTLF, AA, GSH, DTT, CRAT, DCFH and ESR for measurement of OP were reviewed. We compared sensitivity to different PMs species composition, PMs size distribution and seasonality of different assays and concluded that the DTT assay could be the preferred method due to its sensitivity to tracers of combustion derived transition metals and aromatic organic compounds. Specific transition metals (i.e., Mn, Cu, Fe) and quinones made a great contribution to $OP^{DTT}$ value. However, the interactions among PMs components which may change the redox properties of PMs species cannot be ignored, and also insoluble fraction of PMs could not be ignored in assessment of the oxidative potential of PMs.

Additionally, considering that the OP has several dimensions, a combination of two or more OP measures may be needed to create an accurate predictor for health effects.

**Author Contributions:** Conceptualization, L.Z. and X.W.; methodology, T.X.; visualization, S.Z., H.L. and K.X.; writing—original draft preparation, L.R.; writing—review and editing, S.L., X.L., Q.W. and W.W.; All authors have read and agreed to the published version of the manuscript.

**Funding:** This research was funded by the Science and Technology Committee of Shanghai (20ZR1419500); Natural Science Foundation of China (NSFC Grant No. 21477073).

**Acknowledgments:** The authors thank the Science and Technology Committee of Shanghai (20ZR1419500); Natural Science Foundation of China (NSFC Grant No. 21477073) for supporting the research.

**Conflicts of Interest:** The authors declare no conflict of interest.

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
