# Peer review of "Oxidative Potential Induced by Ambient Particulate Matters with Acellular Assays: A Review"

_processes, doi:10.3390/pr8111410_

Round 1
Reviewer 1 Report
I believe that the manuscript has to be strongly revised to make its content more meaningful. Even if several papers are reviewed, the critical comparison among previous studiess is only limited, in particular concerning the still open topics regarding the comparison of acellular oxidative potential to represent health outcomes.

Author Response
Thanks for the comments. We submitted our response to the reviewer in the word file.

Reviewer 2 Report
In this document of Senlin Lu et al, the authors have evaluated the different OP measurement techniques together with their sensitivity to different PMs species composition and size distribution. By comparison of the different acellular assays considered for measurement of OP (RTLF, AA, GSH, DTT, CRAT, DCFH and ESR), the authors concluded that the DTT assay could be the best method due to its sensitivity to tracers of combustion derived transition metals and aromatic organic compounds accumulated in the fine PMs fraction.
The subject is very interesting, the work is well written and offers a clear exposition of the results.
The novelty of the paper is high and the authors succeeded to show the importance of their work given new information / insights they bring with this study.
In my opinion, this manuscript has enough quality to be published in this journal.
Author Response
We appreciated the reviewer's comments and conformation on our manuscript. Thanks so much!
Round 2
Reviewer 1 Report
The manuscripy has been properly revised, by adding some missing information and literature references, according to the Reviewer's suggestion.